# Mitigation of Thermal Instability for Electrical Properties in CaZrO_3_-Modified (Na, K, Li) NbO_3_ Lead-Free Piezoceramics

**DOI:** 10.3390/ma16103720

**Published:** 2023-05-14

**Authors:** Xiaoming Chen, Caoyuan Ai, Zhenghuai Yang, Yuanxian Ni, Xiaodong Yin, Jiankui You, Guorong Li

**Affiliations:** 1School of Materials and Energy Engineering, Guizhou Institute of Technology, Guiyang 550003, China; 2Shanghai Institute of Ceramics, Chinese Academy of Sciences, Shanghai 200050, China

**Keywords:** piezoelectric ceramics, phase transition, thermal instability

## Abstract

Lead-free ceramics 0.96(Na_0.52_K_0.48_)_0.95_Li_0.05_NbO_3_-0.04CaZrO_3_ (NKLN-CZ) are prepared by using the solid-state procedure and two-step synthesis technique. The crystal structure and thermal stability of NKLN-CZ ceramics sintered at 1140–1180 °C are investigated. All the NKLN-CZ ceramics are ABO_3_-type perovskite phases without impure phases. With the increase in sintering temperature, a phase transition occurs in NKLN-CZ ceramics from the orthorhombic (O) phase to the concomitance of O-tetragonal (T) phases. Meanwhile, ceramics become dense because of the presence of liquid phases. In the vicinity of ambient temperature, an O-T phase boundary is obtained above 1160 °C, which triggers the improvement of electrical properties for the samples. The NKLN-CZ ceramics sintered at 1180 °C exhibit optimum electrical performances (*d_33_* = 180 pC/N, *k_p_* = 0.31, dS/dE = 299 pm/V, *ε_r_* = 920.03, tan*δ* = 0.0452, *P_r_* = 18 μC/cm^2^, *T_c_* = 384 °C, *E_c_* = 14 kV/cm). The relaxor behavior of NKLN-CZ ceramics was induced by the introduction of CaZrO3, which may lead to A-site cation disorder and show diffuse phase transition characteristics. Hence, it broadens the temperature range of phase transformation and mitigates thermal instability for piezoelectric properties in NKLN-CZ ceramics. The value of kp for NKLN-CZ ceramics is held at 27.7–31% (variance of *k_p_* < 9%) in the range from −25 to 125 °C. The results indicate that lead-free ceramics NKLN-CZ is one of the hopeful temperature-stable piezoceramics for practical application in electronic devices.

## 1. Introduction

Piezoceramics have diverse applications in numerous fields, such as transducers, sensors, and actuators. But the fact that the most commonly used piezoceramics contain lead and are environmentally harmful during the solid-state reaction process leads to legislation in various countries, e.g., WEEE and RoHS [1,2]. Hence, from an environmental standpoint, the exploitation of green piezoceramics is crucial.

Several lead-free piezoelectric ceramics have undergone extensive research. Among them, (Na,K)NbO_3_ (KNN) is regarded as a promising option for green piezoceramics, given its high piezoelectric constant and Curie temperature (*T*_c_ = 420 °C) [3,4,5]. In 2004, Satio et al. reported that the means of reactive template grain growth was applied in order to prepare KNN-based piezoceramics, which show high electric properties (*d_33_* = 416 pC/N) [6]. Xiao et al. [7] and Zhai et al. [8] have found that in the vicinity of ambient temperature, the ceramics K_0.5_Na_0.5_NbO_3_-CaZrO_3_-LiNbO_3_ form an O-T phase boundary, and the electrical properties can be effectively improved (*d_33_* = 202 pC/N, *T*_c_ = 350 °C). Zhu et al. [9] reported that the Bi_0.5_K_0.5_(Zr, Sn)O_3_-modified (K, Na)(Nb, Sb)O_3_ ceramics have high electric properties (*d_33_* = 460 pC/N, *T*_c_ = 246 °C) through control of the rhombohedral (R)-O-T phase boundary. Zhai et al. [10] prepared CaZrO_3_-modified (K,Na,Li)(Nb,Sb)O_3_ piezoceramics with *d_33_* up to 420 pC/N (*T*_c_ = 214 °C) by combining the formation of the O-T phase boundary with a three-step sintering process. The electric properties of ceramics are greatly improved at room temperature, caused by the construction of phase boundaries, but the Curie temperature (*T*_c_) has significantly decreased, and the performance is highly dependent on temperature. Therefore, the existence of these problems limits their applications.

Over the past few years, numerous researchers have dedicated significant efforts and made some significant breakthroughs in order to vigorously develop KNN-based piezoelectric ceramics with high *d_33_* and *T*_c_, as well as, stable performance. Zhang et al. [11] found that upon lowering the temperature of the O-T phase boundary below 0 °C, the ceramics (Na_0.5_K_0.5_)NbO_3_-LiSbO_3_-CaTiO_3_ exhibit outstanding electric characteristics and a high Curie temperature (*d_33_* = 210 pC/N, *T_c_* = 330 °C), and their performance stability is enhanced simultaneously. The coupling coefficient *k_15_* remains 55.5–55.7% in the range of −50–200 °C. Wang et al. [12] found that when CaZrO_3_ is introduced into (Na, K, Li)(Nb, Ta)O_3_, the ceramics’ rhombohedral (R)-tetragonal phase transition exhibits diffuseness near ambient temperature, and *d_33_*^*^ remains almost unchanged from room temperature to 140 °C. Li et al. [13] found that the constructed phase transformation becomes diffused by doping a certain amount of MnO_2_ into the systems (Na,K)_0.5_NbO_3_-(Bi, Li)_0.5_TiO_3_-BaZrO_3_, and the ceramics not only achieve improved performance (*d_33_*^*^ = 470 pm/V, *T_c_* = 243 °C) but also exhibit good performance stability. *d_33_*^*^ is controlled in the range of 370–470 pm/V from room temperature to 170 °C. Wu et al. [14] achieved a relaxor phase transformation in (K,Na)(Nb,Sb)O_3_-based ceramic systems to obtain excellent electric properties (*d_33_* = 650 pC/N, *T_c_* ≈ 160 °C). Li et al. obtained (Bi, Na) _0.5_HfO_3_-modified (K,Na)_0.5_(Nb,Sb)O_3_ ceramics with excellent electric properties through the establishment of multi-phase coexistence (*d_33_* = 353 pC/N, *T*_C_ = 285 °C) [15]. Moreover, during the preparation process, the optimal sintering temperature is crucial for piezoceramics, which contributes to the formation of dense structures and tailors electric properties. Zhao et al. [16] and Singh et al. [17] optimized the sintering temperature and enhanced the electric properties of potassium sodium niobate-based ceramics. These works indicate that the combination of phase boundary control, diffusion, and sintering effects holds great potential as a promising method for developing KNN-based ceramics with stable performance.

In the previous study, we reported that through constructing a diffused phase boundary, CaZrO_3_ can modify the electric properties in KNN-based piezoceramics [18]. Based on the results, 0.96(Na_0.52_K_0.48_)_0.95_Li_0.05_NbO_3_-0.04CaZrO_3_ piezoceramics are chosen. It mainly focuses on investigating the effect of sintering temperature on the structural evolution and thermal stability of NKLN-CZ ceramics. Moreover, a detailed and systematic analysis of the underlying mechanism is presented.

## 2. Materials and Methods

The 0.96(Na_0.52_K_0.48_)_0.95_Li_0.05_NbO_3_-0.04CaZrO_3_ (NKLN-CZ) piezoceramics were prepared through the solid-state method and two-step synthesis technique. Analytical-grade powders of metal oxides or carbonates, including Na_2_CO_3_ (99.88%), Li_2_CO_3_ (99.31%), K_2_CO_3_ (99.5%), Nb_2_O_5_ (99.96%), CaCO_3_ (99.5%), and ZrO_2_ (99.84%), were employed as initial materials. CaZrO_3_ (CZ) powders were first prepared to obtain the homogeneity of the crystal phase. Analytical-grade metal oxides, ZrO_2_, and carbonate powders, CaCO_3_, were thoroughly blended with anhydrous ethanol and subjected to ball-milling for 6–7 h, followed by calcination at 1300 °C for 2 h. CaZrO_3_ (CZ) powders were obtained after crushing and ball-milling. According to the stoichiometric amount of NKLN-CZ ceramics, all the oxides and carbonates, as well as CZ powders, were ball-milled with anhydrous ethanol for 6–7 h. Following this, the mixed powders underwent drying and were calcined at a temperature range of 800–850 °C for 5–6 h. The reground powders were blended with PVA and subsequently compacted into pellets. Without CZ, NKLN ceramics were sintered at 1120 °C, as shown in Figure 1a. CZ is usually sintered above 1300 °C. It is well known that the range of sintering temperatures for KNN-based ceramics is narrow. Therefore, these temperatures (1140–1180 °C) were chosen to find the optimum *T_sin_* of NKLN-CZ ceramics. They were sintered in an air atmosphere at temperatures ranging from 1140 to 1180 °C for a duration of 3 h. The samples were coated with a silver paste to create electrodes, then annealed at temperatures between 650 and 750 °C. Subsequently, at room temperature, the specimens were poled in a stirred silicone oil bath under a 4–5 kV/mm DC field for a period of 30–50 min.

The crystal phase of samples was measured by X-ray diffraction (Cu Kα radiation, XRD, D8 ADVANCE, Bruker AXS, Karlsruhe, Germany). The density was identified by the Archimedes method [19]. The dielectric properties were obtained through a Novocontrol Alpha-A BroadBand Dielectric Spectrometer (Germany) in the temperature range of −100 ℃ to 250 °C at 3 K/min. The temperature dependence of dielectric properties was obtained by using an impedance analyzer (Agilent, Santa Clara, CA, USA) and a designed furnace from room temperature to 500 °C at 2 °C/min. Through a modified Sawyer–Tower circuit, ferroelectric hysteresis loops (P–E) and electromechanical strain (S–E) were achieved at room temperature (10 Hz, TF Analyzer 2000, aixACCT Systems GmbH, Aachen, Germany). Piezoelectric properties were identified by a Burlincourt-type *d_33_* meter (ZJ-3A, Institute of Acoustic Academia Sinica, Beijing, China). The planar electromechanical coupling factor *k_p_* was identified using Onoe’s formula from frequencies measured through anti-resonance and resonance [20].

## 3. Results and Discussion

### 3.1. Structure

Figure 1 reveals the SEM surface micrographs of NKLN and NKLN-CZ ceramics. All grains exhibit a common square shape. For NKLN ceramics, the average grain size is about 20 μm, as shown in Figure 1a. After the addition of CZ, the grains of NKLN-CZ ceramics remarkably decreased. Meanwhile, with the increase in *T_sin_*, NKLN-CZ ceramics become dense and show abnormal grain growth characteristics. As *T_sin_* is less than 1160 °C, some liquid phases may appear at the grain boundary because of the inhomogeneous existence of alkali metal [21]. When *T_sin_* increases, liquid phases contribute to the growth of abnormal grains and densification of microstructure, which is similar to BaTiO_3_-modified Na_0.5_K_0.5_NbO_3_ ceramics reported by Park et al. [22].

Figure 2a displays XRD patterns of the NKLN-CZ ceramics sintered at temperatures ranging from 1140 to 1180 °C. All of the samples show the ABO_3_-type perovskite structure and are devoid of any impurity phases. It indicates that the compounds are successfully synthesized between the CZ and NKLN compositions. Figure 2b shows the magnified XRD patterns of NKLN-CZ piezoceramics within the 2θ range of 31° to 33° and 44° to 47°. When the sintering temperature (*T_sin_*) is located at 1140–1160 °C, the (111) peak at 2θ = 31–33° can be seen; however, a shoulder peak is formed above 1160 °C. Meanwhile, with the increase in *T_sin_*, the splitting of (202) and (020) peaks at around 2θ of 46° can be observed. The results demonstrate that the sintering effect alters the crystal structure of NKLN-CZ piezoceramics and results in a structural evolution from the O phase to the T phase. This transition should be ascribed to the evaporation of Na and K in the sintering process, consistent with previous reports [21,23].

### 3.2. Dielectric Properties

To further study the structure transition of NKLN-CZ ceramics, the dielectric properties of ceramics within the temperature range of −100 to 200 °C were tested, as shown in Figure 3. Dielectric anomalies have been observed in NKLN-CZ ceramics. The characteristics of these anomalies are analogous to those reported for pure KNN ceramics. Pure KNN ceramics exhibit phase transformation temperatures of O-T (200 °C) and T-cubic (420 °C) [24]. In this case, the structural evolution is easily discernible in NKLN-CZ ceramics. By varying the sintering temperature, a gradual shift is observed in the temperatures of O-T phase transformation (*T_O-T_*) from 150 °C to ambient temperature, as shown in Figure 3a. In order to study the dielectric anomalies, the dielectric loss curves as a function of sintering temperature are selected in Figure 3b. The peaks of the dielectric loss curves from low to high temperatures are used to determine the phase transition of the components: R-O, O-T. With the *T*_sin_ increasing, the O-T phase transition significantly shifts towards low temperature. As a result, the O-T phase boundary appears near room temperature. This phenomenon may be ascribed to the structural changes that originated from the vaporization of sodium and potassium in the sintering process [21]. It is in accord with the XRD findings in Figure 2b. The O–T phase transition of NKLN-CZ ceramics above *T*_sin_ = 1160 °C exhibits proximity to room temperature, which is conducive to generating a large piezoelectric response [25].

In addition, it is noticeable that with the variation in sintering temperature, the *T_c_* (384 °C) of NKLN-CZ ceramics remains nearly constant, as evinced by the data presented in Figure 4. Meanwhile, the relative dielectric constant of NKLN-CZ ceramics exhibits a frequency-dependent behavior, which reflects relaxation characteristics. This phenomenon is in accordance with the modified Curie-Weiss law [26] as follows:$$\frac{1}{\varepsilon_{r}}-\frac{1}{\varepsilon_{r,m}}=C{\left(T-T_{m}\right)}^{\alpha }$$
where *ε_r_* and *ε_r,m_* represent the relative dielectric constants at temperatures *T* and *T_m_*, respectively; *T_m_* is the temperature where the relative dielectric constant gets to a maximum value; *α* is the diffusion coefficient with a value range between 1 and 2; the symbol C represents the Curie–Weiss constant. Figure 5 illustrates the variant of ln(1/*ε_r_* −1/*ε_r,m_*) with respect to ln(*T* − *T_m_*) for NKLN-CZ piezoceramics sintered at temperatures ranging from 1140 to 1180 °C. The slope of spots is achieved by linear regression analysis of the experimental data, and then *α* value can be obtained. The diffuseness coefficients of all samples are in the range of 1 to 2. As we know, when *α* = 1 and 2, the ceramics are categorized as conventional and “complete” diffuse ferroelectrics, respectively. Therefore, the NKLN-CZ ceramics exhibit diffuse ferroelectric behavior. This behavior should be reasonably owed to the disorder in A-site cations caused by the effect of CZ. The cation radius of Ca^2+^ (1.34 Å) approaches that of alkali metal ions (r_Na+_ = 0.98Å, r_K+_ = 1.33Å, r_Li+_ = 1.18Å) [27]. Accordingly, the Ca^2+^ introduction in the A sites of NKLN ceramics can be taken into consideration. When a small number of alkali metal ions evaporate during the high sintering process, Ca^2+^ is able to occupy A-site vacancies in NKLN ceramics and may cause the disorder of A-site cations, which results in a diffuse phase transition [18].

### 3.3. Ferroelectric Characteristics and Electromechanical Strain

Figure 6 reveals the ferroelectric characteristics of NKLN-CZ samples sintered at temperatures between 1140 and 1180 °C. The *P_r_* and *E_c_* display a significant change from 15 μC/cm^2^ and 12.5 kV/cm (*T_sin_* = 1140 °C) to 18 μC/cm^2^ and 14 kV/cm (*T_sin_* = 1180 °C), respectively. The increase in *E_c_* is likely ascribed to the strengthening effect of oxygen vacancies, which is similar to the previous report [21]. During the high sintering process, the alkali metal ions tend to partially evaporate in NKLN-CZ ceramics, which can result in the emergence of vacancies for compensation of ionic charge. The presence of vacancies strengthens the pinning effect and decreases the shift of domains, which results in the growth of *E_c_*. On the other hand, the improved *P_r_* should be owing to the evolution between the O and T phase, which has a positive impact on the electric properties of NKLN-CZ piezoceramics [25].

Figure 7 exhibits the variation in electromechanical strain (S) for NKLN-CZ ceramics sintered at 1140–1180 °C. Similar to the increasing trend of *P_r_*, the maximum electrostrain, which is 0.089%, is achieved by a DC electric field (30 kV/cm). Meanwhile, the value of d*S*/d*E* for NKLN-CZ ceramics achieves 299 pm/V when the sintering temperature is raised to 1180 °C, as illustrated in the inset of Figure 7. Although the increase in electrostrain for NKLN-CZ ceramics may be linked to the vacancies and phase transition, it can be primarily attributed to the following fact: With the increase in sintering temperature, the *T_O-T_* of NKLN-CZ piezoceramics approaches ambient temperature (as seen in Figure 3). Therefore, it is believed that in the vicinity of ambient temperature, the piezoceramics exhibit the potential for structural transition from O to T phase, which contributes to the non-180° domain mobility and generates large electrostrain [28].

### 3.4. Electrical Properties and Thermal Instability

The electrical properties of NKLN-CZ piezoceramics were obtained at room temperature (1 kHz), as illustrated in Table 1. When the sintering temperature increases, the *ε_r_* and tan*δ* display a decreasing trend, obtaining values of 920.03 and 0.0452 at *T*_sin_ = 1180 °C, respectively. Relative density vs. sintering temperature of NKLN-CZ ceramics was given to investigate the effect of *T*_sin_ on structure (see Table 1), which was converted by the theoretical density of ceramics (4.668 g/cm^3^) [21]. The relative density displays a positive correlation with the temperature of sintering, surpassing 95% of its theoretical value beyond 1160 °C. The attainment of a high-density ceramic structure is favorable for improving the electrical properties of NKLN-CZ ceramics. On the other hand, similar to the trend of relative density on sintering temperature, both *d_33_* and *k_p_* exhibit a notable increase, which attains a maximum of 180 pC/N and 0.34, respectively, as sintered at 1160−1180 °C. The improved electric properties of NKLN-CZ ceramics are relevant to density, oxygen vacancies, phase transition, etc., but the main reason can be ascribed to the coexistence of O-T phases. The *T_O-T_* of NKLN-CZ ceramics above 1160 °C is proximal to ambient temperature, as depicted in Figure 3. Therefore, with an increase in *T*_sin_, the ceramics have no difficulty undergoing structural evolution near ambient temperature. It should be responsible for the easy rotation of domains and the production of large piezoelectric responses [25].

To investigate the thermal instability of electric properties for NKLN-CZ ceramics, the curve of *k_p_* vs. *T* for ceramics sintered at 1180 °C is shown in Figure 8. The *k_p_* values of NKLN-CZ ceramics exhibit an initial increase with temperature, followed by a subsequent decrease beyond the peak value. The increase in *k_p_* within the temperature range from −50 to 50 °C can be attributed to the occurrence of an O-T structural transition around ambient temperature, consequently enhancing the electrical performance of NKLN-CZ piezoceramics. However, as the temperature increases, it progressively shifts away from the phase boundary and transitions toward the tetragonal phase, as illustrated in Figure 4. The aforementioned phenomenon results in a decrease in *k_p_*. When the temperature is in the range from −25 to 125 °C, *k_p_* of NKLN-CZ ceramics maintains a range of 27.7–31% (change in *k_p_* < 9%). This phenomenon could potentially be attributed to the underlying mechanism, as elucidated below. When the sintering temperatures change, the NKLN-CZ piezoceramics ultimately form an O-T structural transition in the vicinity of ambient temperature. Due to the close ionic radius, Ca^2+^ may occupy A-site vacancies of NKLN ceramics during the high sintering process, and lead to A-site cations disorder, which exhibits diffuse phase transition characteristics [18]. The appearance of a diffused O-T structural transition results in a broadened temperature range of phase transformation, which contributes to mitigating thermal instability for piezoelectric properties [18,29]. 

## 4. Conclusions

A two-step method was applied to prepare lead-free piezoceramics, NKLN-CZ. The structural evolution and electric performance of NKLN-CZ ceramic systems sintered at 1140–1180 °C have been investigated. The results demonstrate that as the sintering temperature increases, the NKLN-CZ lead-free piezoceramics form a well-defined perovskite without any impurities. The ceramics exhibit a structural transformation from the O phase to a coexisting state of the O-T phases. Through control of the phase boundary, NKLN-CZ piezoceramics sintered at 1180° show good electric performance (*d_33_* = 180 pC/N, *k_p_* = 0.31, d*S*/d*E* = 299 pm/V, *ε_r_* = 920.03, tan*δ* = 0.0452, *P_r_* = 18 μC/cm^2^, *E_c_* = 14 kV/cm) along with a high *T_c_* (384 °C). After CaZrO_3_ enters the NKLN lattice, it may induce disorder among A-site cations and exhibit characteristics of diffuse phase transition, thereby enhancing the relaxation level of systems and broadening the temperature range of phase transformation. At the same time, the ceramic exhibits good temperature stability and electric properties. The *k_p_* value of NKLN-CZ ceramics remains stable within a narrow range of 27.7–31% (a variant of *k_p_* < 9%) over a thermal range of −25 to 125 °C, indicating reliable performance across a wide temperature range. Consequently, this approach offers a means to probe the thermal instability of piezoceramics by tuning the sintering temperature.

## Figures and Tables

**Figure 1 materials-16-03720-f001:**
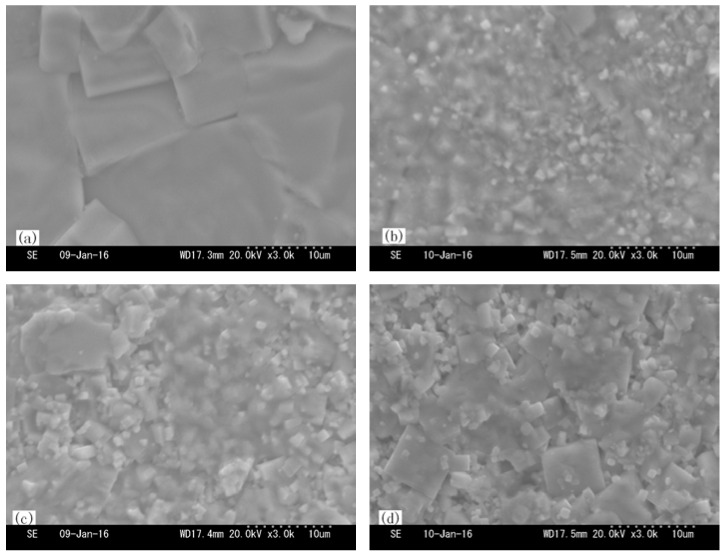
SEM surface micrographs of NKLN. (**a**) *T_sin_* = 1120 °C and NKLN-CZ ceramics; (**b**) *T_sin_* = 1140 °C; (**c**) *T_sin_* = 1160 °C; (**d**) *T_sin_* = 1180 °C.

**Figure 2 materials-16-03720-f002:**
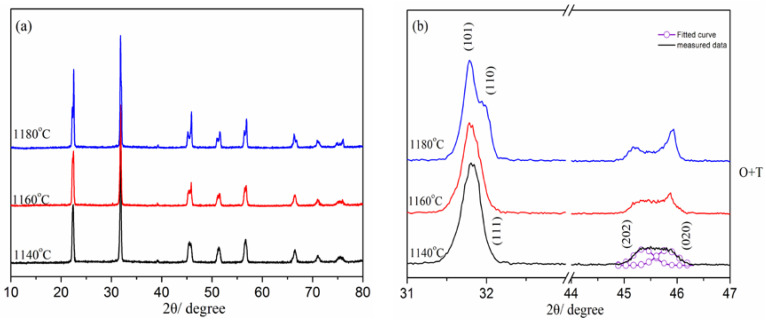
(**a**) XRD patterns of NKLN-CZ piezoceramics sintered at 1140–1180 °C; (**b**) The expanded XRD patterns of NKLN-CZ piezoceramics in the range of 31–47°.

**Figure 3 materials-16-03720-f003:**
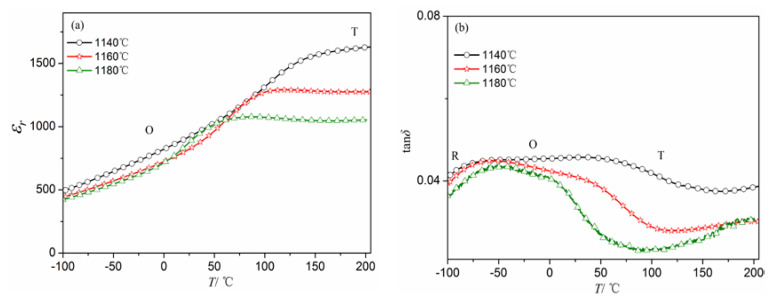
Temperature dependence of *ε_r_* (**a**) and tan*δ* (**b**) for NKLN-CZ piezoceramics sintered at different temperatures (10 kHz).

**Figure 4 materials-16-03720-f004:**
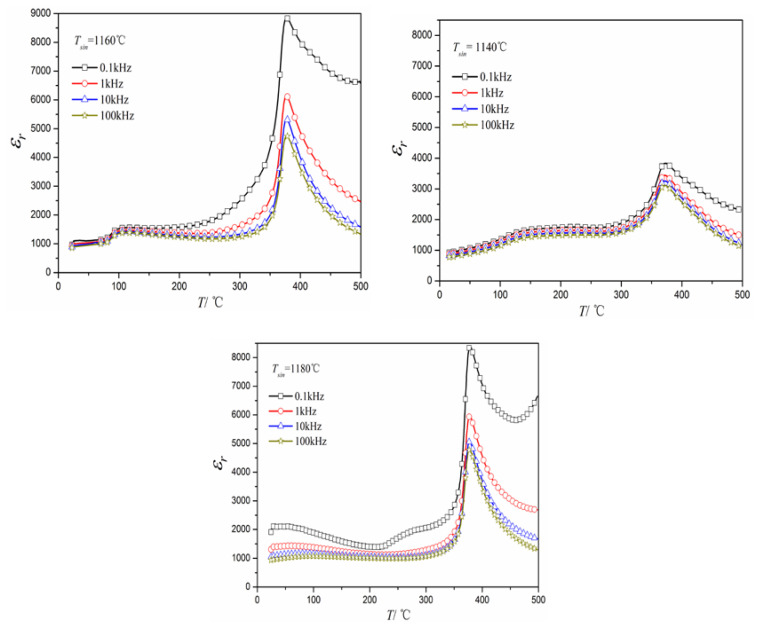
Temperature dependence of dielectric properties for NKLN-CZ piezoceramics sintered at 1160–1180 °C.

**Figure 5 materials-16-03720-f005:**
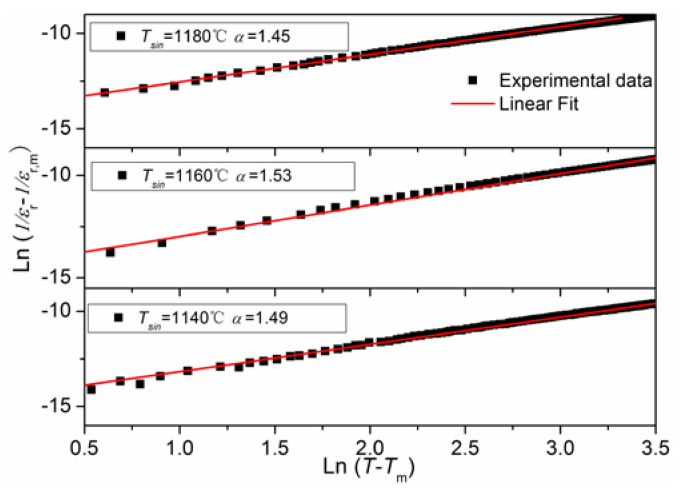
ln(1/*ε_r_* − 1/*ε*_r,m_) vs. ln(*T* − *T_m_*) for the NKLN-xCZ piezoceramics (10 kHz).

**Figure 6 materials-16-03720-f006:**
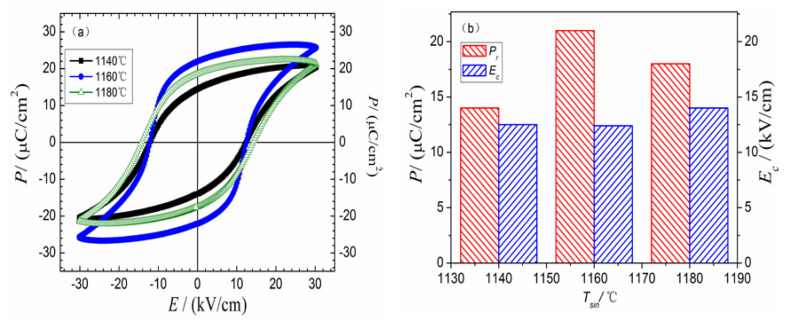
*P*–*E* loops (**a**) and *P_r_*(*E_c_*) vs. *T_sin_* (**b**) for the NKLN-CZ ceramics (10 Hz).

**Figure 7 materials-16-03720-f007:**
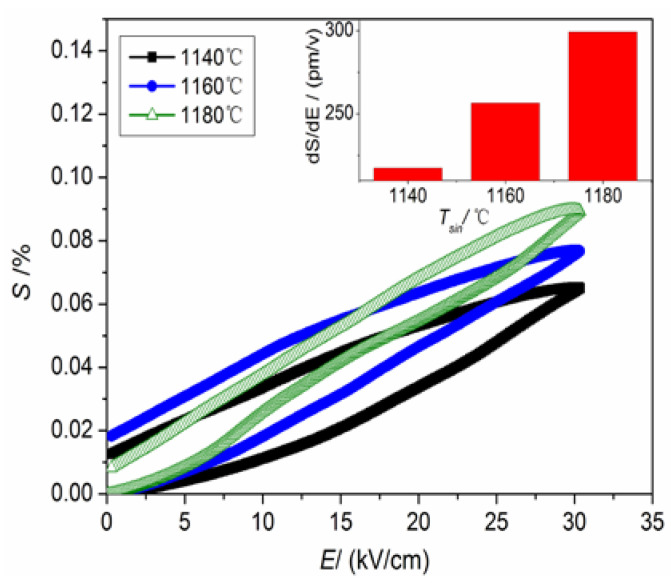
Electromechanical strain for NKLN-CZ ceramics sintered at 1140–1180 °C.

**Figure 8 materials-16-03720-f008:**
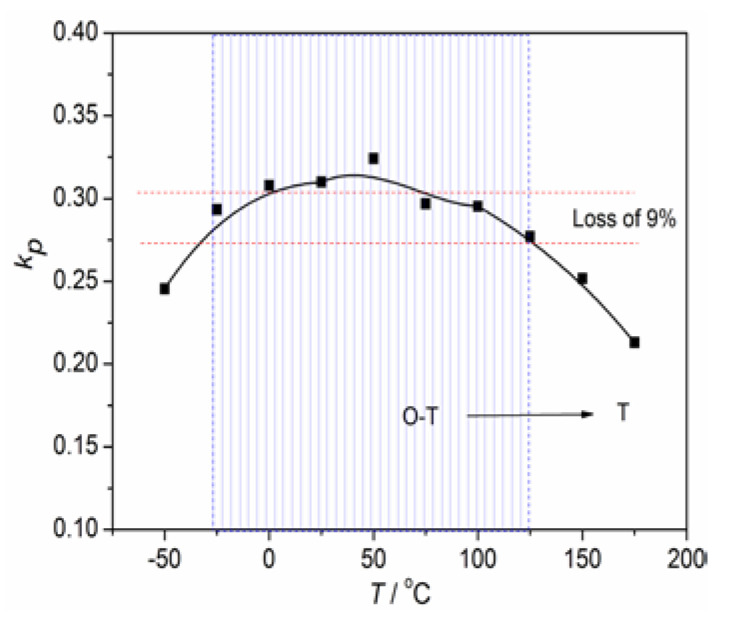
The temperature vs. electrical properties for NKLN-CZ piezoceramics.

**Table 1 materials-16-03720-t001:** Electric properties of NKLN-CZ ceramics sintered at 1140–1180 °C.

*T*_sin_ (°C)	*d_33_* (pC/CN)	*k_p_*	*ε_r_*	tan*δ*	Relative Density (%)
1140	96	0.28	987.85	0.0561	94.88
1160	159	0.34	873.43	0.0408	95.52
1180	180	0.31	920.03	0.0452	95.18

## Data Availability

Data is available from the corresponding author upon reasonable request.

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
