# Peer review of "Mitigation of Thermal Instability for Electrical Properties in CaZrO3-Modified (Na, K, Li) NbO3 Lead-Free Piezoceramics"

_materials, 2023, doi:10.3390/ma16103720_

Round 1

Reviewer 1 Report

Comments on the manuscript "Mitigation of thermal instability for electrical properties in CaZrO3-modified (Na, K, Li) NbO3 lead-free piezoceramics". In this work, the authors explore the effect of sintering temperature on the ferroelectric properties of 0.96(Na0.52K0.48)0.95Li0.05NbO3-0.04CaZrO3 (NKLN-CZ). Additionally, the study focuses on enhancing the ferroelectric properties by optimizing the sintering conditions. The samples are characterized using XRD, LCR, and hysteresis, and the data is well explained. Overall, the topic is interesting and suitable for publication in the Materials journal. I have the following comments, which should be considered before accepting the manuscript for publication.

·     -    It is necessary to include electron microscopy images of each sample in the manuscript. Additionally, the authors should determine the grain size distribution in the SEM figure.

·      -   The authors should explain why these temperatures were chosen.

·       -  Finally, the hysteresis curve of the sample sintered at 1180°C should be redone since it does not match the obtained values.

Reviewer 2 Report

Comment 1: In the abstract, the authors state that the O-T phase boundary triggers the improvement of density. In the revised manuscript, please explain how the O-T boundary can improve the density of the samples.

Comment 2: In the abstract, the authors say that the ceramics show relaxor behaviour. I think this should be changed to diffuse phase transition behaviour, as the value of Tc does not change with measurement frequency.

Comment 3: The microstructure and grain size in particular will have an effect on the dielectric and piezoelectric properties, but this is not mentioned in the manuscript. Please include some SEM micrographs of the microstructure of the samples, and discuss how microstructure changes with sintering conditions.

Comment 4: In section 3.1 structure, the authors fit peaks to one of the XRD patterns, but they do not discuss this in the results. Please discuss the fitted peaks in the revised manuscript. Also, please do peak fitting on the other samples. Please include figures of the Rietveld refinement of the XRD patterns in the revised manuscript. The sample sintered at 1160°C appears to show both orthorhombic and tetragonal phases. Why was Rietveld refinement not carried out on the sample sintered at 1160°C? The sample sintered at 1180°C appears to have completely changed to tetragonal, but Table 1 indicates that it is still a mixture of orthorhombic and tetragonal. Is this correct?

Comment 5: In Figure 2, loss tangent decreases as sintering temperature increases. I would have expected loss tangent to increase, as increasing alkali evaporation would increase the number of vacancies in the crystal lattice. Is the decrease in loss tangent related to the decrease of the O/T phase transition temperature? Please discuss this in the revised manuscript.

Comment 6: Please define Tm in the modified Curie-Weiss law in the revised manuscript.

Comment 7: Please check the attached file for some corrections to the text.

Reviewer 3 Report

The article discusses the properties of 0.96(Na0.52K0.48)0.95Li0.05NbO3-0.04CaZrO3(NKLN-CZ) ceramics obtained by solid-phase synthesis, and establishes the relationship between the composition of ceramics and their thermal stability and electrical properties depending on sintering conditions. The authors used a fairly large number of different research methods, which allowed the authors to obtain a number of new dependencies, as well as to establish new data on structural properties. In general, this work has a number of unique results, which makes it promising for this journal. However, before the article is accepted for publication, the authors should answer a number of questions from the reviewer.

1. The authors should give explanations regarding the choice of the annealing temperature range, with a small step. With such small changes in annealing temperatures, the authors should provide more details on exactly how the heating was controlled.

2. The authors should explain exactly how the CaZrO3 were introduced, leading to a change in the properties of the materials.

3. X-ray phase analysis of the studied ceramics should be presented with a more detailed analysis, including a detailed analysis of the distortions of the crystal structure associated with changes in sintering conditions.

4. Structural parameters should be given with a measurement error, and explanations should be given for the presence of two phases in the structure of ceramics.

5. Dielectric properties should be given taking into account the analysis of structural changes, as well as changes in the density of ceramics.
